Antimicrobial consumption on Austrian dairy farms: an observational study of udder disease treatments based on veterinary medication records

Firth Clair L. clair.firth@vetmeduni.ac.at 1
Käsbohrer Annemarie 1
Schleicher Corina 2
Fuchs Klemens 2
Egger-Danner Christa 3
Mayerhofer Martin 3
Schobesberger Hermann 1
Köfer Josef 1
Obritzhauser Walter 1
1 Institute of Veterinary Public Health, University of Veterinary Medicine , Vienna , Austria
2 Data, Statistics and Risk Assessment, Austrian Agency for Health and Food Safety (AGES) , Graz , Austria
3 ZuchtData EDV-Dienstleistungen GmbH , Vienna , Austria
Byrne Andrew
Electronic publication date: 2017 Nov 16
Publication date: 2017
Volume: 5
Electronic Location ID: e4072
Received 2017 Aug 29; Accepted 2017 Oct 30
Copyright: ©2017 Firth et al.
Copyright year: 2017
Copyright holder: Firth et al.
License: This is an open access article distributed under the terms of the Creative Commons Attribution License, which permits unrestricted use, distribution, reproduction and adaptation in any medium and for any purpose provided that it is properly attributed. For attribution, the original author(s), title, publication source (PeerJ) and either DOI or URL of the article must be cited.
License URL: https://creativecommons.org/licenses/by/4.0/

Keywords: Antimicrobial, Antibiotic, Dairy cow, Mastitis, DDD, Defined daily dose, Udder disease, Cattle

Funding: Austrian Ministry for Transport, Innovation and Technology (BMVIT) Federal Ministry of Science, Research and Economy (BMWFJ) Austrian Research Promotion Agency (FFG) Austrian Federal Ministry of Health and Women’s Affairs (BMGF) The K-Projekt ADDA—Advancement of Dairying in Austria was supported by the Austrian Ministry for Transport, Innovation and Technology (BMVIT), the Federal Ministry of Science, Research and Economy (BMWFJ), the province of Lower Austria and the city of Vienna within the framework of COMET—Competence Centers for Excellent Technologies. The COMET program is handled by the Austrian Research Promotion Agency (FFG). The electronic infrastructure to record harmonised treatment data was supported by the project “Electronic Herdbook”, funded by the Austrian Federal Ministry of Health and Women’s Affairs (BMGF). The funders had no role in study design, data collection and analysis, decision to publish, or preparation of the manuscript.

==============================
Background

Antimicrobial use in livestock production is an important contemporary issue, which is of public interest worldwide. Antimicrobials are not freely available to Austrian farmers and can only be administered to livestock by veterinarians, or by farmers who are trained members of the Animal Health Service. Since 2015, veterinarians have been required by law to report antimicrobials dispensed to farmers for use in food-producing animals. The study presented here went further than the statutory framework, and collected data on antimicrobials dispensed to farmers and those administered by veterinarians.

Methods

Seventeen veterinary practices were enrolled in the study via convenience sampling. These veterinarians were asked to contact interested dairy farmers regarding participation in the study (respondent-driven sampling). Data were collected from veterinary practice software between 1st October 2015 and 30th September 2016. Electronic data (89.4%) were transferred via an online interface and paper records (10.6%) were entered by the authors. Antimicrobial treatments with respect to udder disease were analysed by number of defined daily doses per cow and year (nDDDvet/cow/year), based on the European Medicines Agency technical unit, Defined Daily Dose for animals (DDDvet). Descriptive statistics and the Wilcoxon rank sum test were used to analyse the results.

Results

Antimicrobial use data from a total of 248 dairy farms were collected during the study, 232 of these farms treated cows with antibiotics; dry cow therapy was excluded from the current analysis. The mean number of DDDvet/cow/year for the antimicrobial treatment of all udder disease was 1.33 DDDvet/cow/year. Of these treatments, 0.73 DDDvet/cow/year were classed as highest priority critically important antimicrobials (HPCIAs), according to the World Health Organization (WHO) definition. The Wilcoxon rank sum test determined a statistically significant difference between the median number of DDDvet/cow/year for acute and chronic mastitis treatment (W = 10,734, p < 0.001). The most commonly administered antimicrobial class for the treatment of acute mastitis was beta-lactams. Intramammary penicillin was used at a mean of 0.63 DDDvet/cow/year, followed by the third generation cephalosporin, cefoperazone, (a HPCIA) at 0.60 DDDvet/cow/year. Systemic antimicrobial treatments were used at a lower overall level than intramammary treatments for acute mastitis.

Discussion

This study demonstrated that Austrian dairy cows in the study population were treated with antimicrobial substances for udder diseases at a relatively low frequency, however, a substantial proportion of these treatments were with substances considered critically important for human health. While it is vital that sick cows are treated, reductions in the overall use of antimicrobials, and critically important substances in particular, are still possible.

Introduction

Antimicrobial consumption (AMC) in food-producing animals is a contemporary topic of particular relevance to veterinarians, farmers, authorities, public health professionals and the general public worldwide, which attracts considerable media and political interest. Although the treatment of clinical disease is vital for animal welfare and food safety, food producers, restaurants and supermarkets are increasingly marketing their products as being “raised without antibiotics” and there is an expectation among the general public that antimicrobial use should be reduced (NRDC, 2015; Smith, 2015). While the average consumer perceives such marketing strategies as a positive development for food quality, the likelihood of increased suffering among sick animals left untreated cannot be ignored. Though the link between antimicrobial resistance (AMR) in livestock and humans remains unclear (Deiters et al., 2015; Schmidt, Kock & Ehlers, 2015; Walter et al., 2017), a number of studies have highlighted the possible relationship between AMC on farm and the development of AMR (Garcia-Migura et al., 2014; Hille et al., 2017) and, as such, the use of antimicrobials in farming is coming under increasing scrutiny. In particular, there have been calls for antimicrobials classed by the World Health Organization (WHO) as “highest priority critically important” in human medicine (namely macrolides, fluoroquinolones, third and fourth generation cephalosporins, and polymyxins) to be restricted for use as veterinary medicinal products (O’Neill, 2016; World Health Organization, 2017).

In Austria, veterinary medicines are strictly regulated by law and antimicrobial agents are not permitted to be sold over-the-counter (BMGF, 2002). Veterinarians and farmers have been required to complete and retain statutory documentation on each livestock treatment for over a decade, in accordance with both local and European Union legislation. Furthermore, veterinarians are only permitted to dispense certain antimicrobial agents if the farmers are members of the Animal Health Service (German: Tiergesundheitsdienst—TGD) and have been adequately trained in the administration of intramammary, intramuscular and subcutaneous medicinal products (BMG, 2009). If farmers are not members of this health service system, then veterinarians are not permitted to leave any antimicrobial agents on farm (with the exception of oral preparations), and all such products, including intramammary applications, must be administered to livestock by a veterinarian. The reporting of antimicrobial dispensing (per farm) by veterinary practices to the relevant authorities in Austria has been a legal requirement since 1st January 2015 (BMG, 2014). Veterinarians treating food-producing animals are required to report all antimicrobial agents dispensed to farmers, but not those that they administer to livestock themselves.

In order to develop effective recommendations to reduce antimicrobial use in livestock, it is first necessary to assess the current level of veterinary antimicrobial consumption at the level of both the animal species and the production type. Since 2009, the EU has collected sales data from wholesale pharmacies and pharmaceutical companies and published these in the European Surveillance of Veterinary Antimicrobial Consumption (ESVAC) report, covering data from 29 EU/EEA countries in its most recent version (EMA, 2016a). Similarly, in the US, the Food and Drug Administration collates data on antimicrobial agent sales for use in food-producing animals (FDA, 2016). Worldwide, the situation is harder to assess, as only 42 of 154 World Organisation for Animal Health (OIE) member countries were reported in 2014 to be gathering data on antimicrobial consumption in food-producing animals and a more recent survey in 2017 determined that 56 of 134 OIE members were actively collecting and analysing AMC data (Rushton, Ferreira & Stärk, 2014; FAO, 2016; Yahia, 2017). While wholesale data are useful for gathering information on treatment trends, they can never replace actual treatment data as they fail to include species treated or the relevant production system. However, the European Commission, the United Nations, the World Health Organization and the OIE have all recently published action plans pledging to improve the level of monitoring of AMC and to combat antimicrobial resistance on a global scale (World Health Organization, 2015; OIE, 2016; Thomson, 2016; EC, 2017).

Comparing antimicrobial consumption data between countries is challenging, as monitoring systems vary greatly (Merle et al., 2012; DANMAP, 2016). Furthermore, a variety of units has been used to calculate daily doses or the amount of active substance used per epidemiological unit (e.g., animal/liveweight/“population correction unit” etc.) (Grave et al., 1999; Chauvin et al., 2001; EMA, 2013; EMA, 2016b; Bondt et al., 2013; Postma et al., 2015). At times, these units have been defined by individual countries (e.g., Denmark, the Netherlands) and are often not applicable to other data collection systems. In hospitals and public health epidemiology, “Defined Daily Doses” (DDD) have been used for many years to analyse medication use in humans (Chauvin et al., 2001; Collineau et al., 2017). In April 2016, the European Medicines Agency (EMA) published the first standardised “Defined Daily Doses” for veterinary medicine (DDDvet) which have been used in this study in an attempt to describe actual antimicrobial use with a European standardised technical unit, enabling an assessment of AMC for each active substance (EMA, 2016b). Although complex, it is important to establish the baseline level of AMC on dairy farms prior to interventions so that veterinarians and farmers can then work together to decrease disease incidence, subsequently reducing the level of AMC, while maintaining a high level of milk quality and animal welfare (Reyher, 2016).

The study presented here collated electronic AMC data from the practice management software of 17 veterinary practices, covering a total of 248 dairy farms and the equivalent of 6,467 cow-years. The aim of the study was to determine the level of antimicrobial use for the treatment of udder disease, as prescribed by herd veterinarians, on these Austrian farms over a one-year observational period.

Materials and Methods

Study population

Veterinary treatment data were collected as part of a larger interdisciplinary project entitled “ADDA—Advancement of Dairying in Austria”, which included both academic and industry partners (e.g., cattle breeding associations, milk processors, farmers’ unions). A convenience sample of veterinary practices primarily involved in dairy production medicine was contacted directly by the authors. Seventeen veterinary practices (29 veterinarians) provided AMC data in this study. The number of veterinarians per practice ranged from one to five (mean 1.7, median 1), with 13 practices being run by a single practitioner. Almost half of the practices treated only farm animals (46.7%), with the remainder (53.3%) being mixed practices (i.e., treating both farm and small animals).

Enrolled veterinarians actively recruited interested farmers from their catchment areas (respondent-driven sampling). As such, this was not a randomised sample of veterinarians and dairy farmers throughout Austria, but was a convenience sample within an observational study. The authors did not influence the veterinarians’ choice of farmer. No restrictions were made with respect to farm size, production system, freestalls or tie-stalls, alpine or valley farms, etc. However, all veterinarians and dairy farmers were members of the Austrian Animal Health Service, and the dairy farmers also participated in the national milk-recording scheme (German: Landeskontrollverband, LKV). The study was discussed and approved by the institutional ethics and animal welfare committee of the University of Veterinary Medicine, Vienna (Ref No. ETK-13/11/2015), in accordance with good scientific practice (GSP) guidelines and national legislation. A total of 283 dairy farmers in the federal states of Lower Austria, Upper Austria, Styria, Carinthia and Tirol initially agreed to take part in the study; 253 of these signed an informed consent form and approved the use of data from their farms for research purposes within the context of this study.

Data collection

The data collection period with respect to antimicrobial treatments began on 1st October 2015 and concluded on 30th September 2016. An online interface was created, using the Electronic Herdbook program, to which the respective veterinarians could upload AMC data from their practice management software. To simplify the process for participating veterinarians, electronic veterinary treatment data for all bovine animals (cows, calves, youngstock) were collected. Table 1 illustrates the type of electronic data collected from veterinarians with respect to each treatment. As well as the farm identification number, an animal identification (ear tag) number was entered for each animal, meaning that analysis at both herd and animal level was possible. Production type for the treated animal was also recorded as the small-scale structure of Austrian dairy farms means that some male calves/youngstock are often kept on dairy farms for fattening.

Table 1 Data collected from veterinarians’ electronic medication records based on the interface of the Electronic Herdbook program.

Data	Explanation	
Veterinarian ID	Refers to each individual veterinarian	
Veterinary pharmacy ID	Refers to each practice	
Internal reference no.	Relevant only to individual practice software	
Receipt number		
Farm ID	Identifies each individual farm	
Treatment date		
Animal ID	Ear tag number	
Production type	e.g., dairy, beefa	
Number of treated animals		
Diagnosis code	Legal requirement in Austria. Codes available for most disorders e.g., 51 = acute mastitis, others coded as “not otherwise specified” (NOS)	
Indication	If information additional to diagnosis code was entered by the veterinarian (optional)	
Continuation of previous treatment?	Y/N	
Medication licensing number		
Medication lot number		
Administration or dispensing?	Administration = treatment carried out by veterinarian. Dispensing = medication left on farm for farmer to administer at a later date, according to veterinary instructions	
Medication amount	e.g., 20, 1, 4	
Medication unit	e.g., ml, unit, litre	
Dosage		
Recommended dosage		
Length of treatment	In days	
Administrations per day		
Mode of application	e.g., intramammary, intramuscular	
Instructions for application (to be printed on label of dispensed medication)	e.g., 6 ml on 3–4 consecutive days	
Statutory withdrawal period (milk)	In days	
Statutory withdrawal period (meat)	In days	
Animal species	Cattle	
Notes.

a NB: All farms included in this analysis were dairy farms, however, given the small structure of Austrian agricultural systems, many farms retain some male calves/youngstock on farm for fattening/beef production.

These data were then validated based on animal and farm identification data from the central cattle database (German: Rinderdatenverbund, RDV system), and a comparison of medication licensing number with the database of the Austrian Medicines and Medical Devices Agency. Antimicrobial use data were then extracted from the dataset into a Microsoft (Microsoft Corporation, Redmond, WA, USA) Excel spreadsheet for further analysis. Statistical analysis was carried out using Microsoft Excel and the R software package, Version 3.3.2 (R Core Team, 2016). Where necessary, data were given pseudonyms (according to local statutory requirements) so that neither the veterinarian nor the farm was identifiable.

According to an Austrian law passed in 2002, diagnoses must be documented for any veterinary medication dispensed or administered to livestock (BMGF, 2002). Diagnoses were reported by the herd veterinarian according to the official cattle disease diagnosis code system (as defined by the Austrian Federal Ministry of Health in 2006), which includes more than 60 possible clinical diagnoses (plus “not otherwise specified” for complexes which do not fit into other listed categories) (Egger-Danner et al., 2012). With respect to udder disease, possible diagnoses included “acute mastitis”, “chronic mastitis” or “other udder disorders” (including disorders of the teat or udder skin, udder oedema, teat injuries etc.). (NB. The current analysis did not include prophylactic drying off). Diagnoses were assigned by the herd veterinarian according to his/her clinical assessment of the animal, no additional external criteria were applied.

If electronic data were incomplete or the veterinary practice software proved incompatible with the online interface, data were provided by the participating veterinarians as paper records, which were then entered into a Microsoft Access database by one of the authors. This database was then exported into a Microsoft Excel spreadsheet and combined with the electronic treatment records for further analysis.

Antimicrobial consumption data

AMC data were analysed by diagnosis code (including all codes relating to udder disease, but excluding dry cow therapy), mode of application (e.g., intramammary instillation, injection) and whether the substances used were classed as “highest priority critically important antimicrobials” (HPCIAs) (World Health Organization, 2017). Further analyses were done by antimicrobial class, herd and per veterinary practice.

Amounts of antimicrobial active substance were calculated by multiplying the volume of veterinary medicinal product administered to the animal (usually given in millilitres [ml]) by the concentration of the antimicrobial active substance (e.g., mg/ml) to give the total mass of antimicrobial active substance in milligrams (mg). The number of DDDvet administered were calculated using the DDDvet values assigned to the individual antimicrobial substances and animal species by the European Medicines Agency (EMA, 2016b) as follows:

nDDDvet=amount active substance (mg)DDDvet for that antimicrobial active substance

To calculate the nDDDvet/cow/year, the following formula was used:

nDDDvet∕cow/year=nDDDvetProduction days×Standardised liveweight (500 kg)×365

Only cows were included in the analysis of antimicrobial use for udder disease treatments. The standardised liveweight of 500 kg for a dairy cow was taken from European Medicines Agency guidelines (EMA, 2013). Production days were defined as the actual number of days an animal was kept in the herd during the observational period (according to livestock movement data from the central cattle database). The number of cow-years per herd was calculated using the total number of cow production days per herd divided by 365.

Intramammary treatment injectors were classed as “one treatment per tube”. In accordance with the EMA definitions for DDDvet for intramammary treatments, where the defined daily dose is given as one treatment per teat (EMA, 2016b), no calculations were necessary with respect to the amount of active substance. The DDDvet for intramammary treatments as defined by the EMA is based on daily dose per cow, rather than per kilogram liveweight. The calculation of the DDDvet/cow/year for intramammary treatments did not, therefore, include the standardised liveweight of the cows treated. The number of production days per farm, based on animal movement data, was, however, always included in the number of daily doses per cow and year.

The number of defined daily doses per cow and year for intramammary therapies was calculated as follows:

nDDDvet∕cow/year(intramam.)=nDDDvetProduction days×365

Oxytetracycline sprays for the local treatment of udder disorders are not included in this analysis as no DDDvet value has been assigned by the EMA to antimicrobial sprays. This decision led to the exclusion of six spray cans of oxytetracycline dispensed over the one-year period. Similarly, dry cow treatments (DCT) are not included here, as no DDDvet values have been allocated to these products by the European authorities (EMA, 2015; EMA, 2016b). Calculation of antimicrobial treatment with dry cow injectors is instead standardised by the EMA using the Defined Course Dose (DCDvet), which is classified as four treatments per udder. To avoid confusion with DDDvet values, it was decided not to include DCT in the current analysis, which concentrated on the frequency of antimicrobial treatment of mastitis and other udder diseases, rather than the primarily prophylactic use of antimicrobials at drying off.

Statistical analysis

The total number of DDDvet/cow/year was calculated for each herd and the mean, standard error of the mean (SEM) and median of the herd values were then determined for each udder disease diagnosis code as assigned by the treating veterinarian. The analysis was based on the treated population of dairy cows, however, descriptive statistics were calculated only for those herds where animals were treated for a particular diagnosis. The number of treatments calculated on the basis of defined daily doses are presented as boxplots, illustrating the minimum, maximum, mean, median, 25th and 75th percentiles and outliers of DDDvet/cow/year values.

The median DDDvet/cow/year values for acute and chronic mastitis treatments were compared using a non-parametric statistical test, the Wilcoxon rank sum test with continuity correction for unpaired samples. The diagnosis group “other udder disorders” was not included in the statistical analysis as the number of cases reported was too small to allow for a meaningful analysis.

Results

Farm demographics

The number of herds enrolled (and whose herd veterinarians provided AMC data) in the study is described in Fig. 1. Based on the 253 herds where farmers provided informed consent, herd size ranged from nine to 240 head of cattle (mean 55.6; median 45), with numbers of dairy cows ranging from five to 98 (mean 26.2; median 21). The vast majority (73.8%) of cows were of the Simmental (Austrian Fleckvieh) breed, with the remainder made up of Brown Swiss (Braunvieh, 10.9%) and Holstein-Friesians (12.7%). Approximately two-thirds (65.5%) of farms kept their cows in freestalls, primarily cubicles or straw-yards, whereas 34.5% used a tie-stall system. As a consequence of these housing systems, just over half (52.6%) of the farms had milking parlours, with 32.9% milking via a vacuum-line in the barn. Twelve farms (4.8%) used automated milking systems and a small number (4%) still milked directly into a bucket.

Figure 1 Flowchart to describe the study population with respect to participating herds and AMC data collection.

AMC, antimicrobial consumption; DCT, dry cow therapy.

Veterinary medication records

Medication data on all administered and dispensed antimicrobial products were collected electronically for 248 enrolled dairy farms from the practice management software systems of 17 veterinary practices (Fig. 1). Where electronic AMC data were incomplete, paper records were included in the dataset. This was necessary for a total of 48 (19.4%) farms and made up 10.6% of total treatment data.

Overall, 6,334 single antimicrobial treatments of dairy cows were collated during the one-year observational study period.

Reported diagnoses for udder disease

Once calves and youngstock were excluded, 232 herds were determined to have treated dairy cows with antimicrobial agents at least once over the one-year study period (for details see Fig. 1). The most commonly reported initial diagnosis treated with antimicrobial substances within the group of “udder diseases” over the one-year period was acute mastitis (Table 2). Diagnoses are reported as designated by the prescribing veterinarian.

Table 2 Number of reported initial diagnoses for udder disease and number of data entries for diagnoses treated with antimicrobial substance.

Reported diagnosis code	Number (%) of initial diagnoses	Number (%) of data entries for diagnoses with antimicrobial treatmenta	
Acute mastitis	1,194 (75.9)	2,016 (87.5)	
Chronic mastitis	208 (13.2)	260 (11.3)	
Other udder disorders	171 (10.9)	28 (1.2)	
Total udder diseases	1,573	2,304	
Notes.

a If two or more different antimicrobial substances were used to treat udder disease, then the diagnosis code was entered into the system each time.

Table 3 Descriptive statistics of the number of Defined Daily Doses (DDDvet) per cow and year for antimicrobials used to treat udder disease, based on European Medicines Agency technical units.

Study population: 232 dairy herds where cows were treated with antimicrobials during a one-year observational period.

	Herds where treatment necessary	% of 232	nDDDvet/cow/year	
			Mean	SEM	Median	Minimum	Maximum	
All udder diseasea	
Total AMC	214	92%	1.33	0.09	0.86	0.04	9.99	
HPCIA	183	79%	0.73	0.08	0.41	0.03	9.67	
Acute mastitis	
Total AMC	205	88%	1.21	0.09	0.71	0.03	9.99	
HPCIA	175	75%	0.71	0.08	0.36	0.03	9.67	
Chronic mastitis	
Total AMC	72	31%	0.48	0.05	0.34	0.02	2.55	
HPCIA	40	17%	0.25	0.04	0.16	0.03	1.05	
Notes.

a Some farms may have reported cases of both acute and chronic mastitis over the study period, meaning that the sum of farms reporting acute and chronic mastitis exceeds the total number of farms with udder disease.

nDDDvet/cow/year number of defined daily doses per cow and year

SEM standard error of the mean

AMC antimicrobial consumption

HPCIA highest priority critically important antimicrobial

Figure 2 The distribution of the number of Defined Daily Doses (DDDvet) per cow and year by individual herd for all reported diagnoses of “udder disease”—for total AMC and for HPCIA.

AMC, antimicrobial consumption; HPCIA, highest priority critically important antimicrobials. Box, the range between the 1st (Q1) and 3rd (Q3) quartile; horizontal line, median; x, mean; lower whisker, Q1−1.5(IQR) (interquartile range); upper whisker, Q3 + 1.5(IQR); dots, outliers.

Figure 3 The distribution of the number of Defined Daily Doses (DDDvet) per cow and year by individual herd for reported diagnoses of (A) acute mastitis and (B) chronic mastitis—for total AMC and for HPCIA.

AMC, antimicrobial consumption; HPCIA, highest priority critically important antimicrobials. Box, the range between the 1st (Q1) and 3rd (Q3) quartile; horizontal line, median; x, mean; lower whisker, Q1−1.5(IQR) (interquartile range); upper whisker, Q3 + 1.5(IQR); dots, outliers.

Antimicrobial consumption

A total of 2,304 antimicrobial treatments were designated as being for udder diseases, this group made up 36.4% (2,304/6,334) of all antimicrobial treatments in this study population. Overall, 92% (214/232) of all herds where antimicrobial substances were administered to lactating cows treated these animals for “udder disease”, with 79% of all farms using HPCIA for these treatments (Table 3). Concentrating the analysis solely on these 214 herds where cows were treated for udder disease, the mean DDDvet/cow/year was determined to be 1.33 (±0.09 SEM) for all udder disease treatments, with a median of 0.86, ranging from 0.04 to 9.99 DDDvet/cow/year (Table 3 and Fig. 2). Cows suffering from acute mastitis were reported to have been treated on 205 farms, with a mean DDDvet/cow/year of 1.21 (±0.09) for all antimicrobial consumption, and a mean DDDvet/cow/year of 0.71 (±0.08) for HPCIAs (Fig. 3A and Table 3). Chronic mastitis cases were treated much less frequently, on only 72 farms, with a mean DDDvet/cow/year of 0.48 (±0.05) for all antimicrobial consumption and a mean DDDvet/cow/year of 0.25 (±0.04) for HPCIAs (Fig. 3B and Table 3).

The highest level of use by antimicrobial class was determined to be penicillins at a mean of 0.16 (±0.02) DDDvet/cow/year administered systemically and 0.63 (±0.08) DDDvet/cow/year for intramammary use (Table 4). With respect to the maximum level of use on individual farms, however, the fourth generation cephalosporin, cefquinome, was reported to be administered at the highest level for both intramammary and systemic use (ranging from 0.04 to 6.65 DDDvet/cow/year and 0.01 to 1.11 DDDvet/cow/year, respectively) (Table 4).

The Wilcoxon rank sum test determined that the difference between the medians of the herd level DDDvet/cow/year with respect to acute mastitis and chronic mastitis treatments was significant at the 5% level, (acute mastitis n = 205: median 0.71, 95% CI [0.61–0.89] and chronic mastitis n = 72: median 0.34, 95% CI [0.23–0.52]), with a test statistic W = 10,734, p-value < 0.001.

Treatments by veterinary practice

Antimicrobial treatments for udder disease by veterinary practice are shown in Fig. 4. To ensure the anonymity of practices participating in this study, the number of veterinarians per practice is not shown. The proportion of defined daily doses by cow and year to treat udder disease with non-critical antimicrobial substances varied considerably between veterinary practices, with the use of HPCIAs extending from a minimum of 8% of all defined daily doses for antimicrobial substances administered and/or dispensed on the study farms by practice #502 to >80% HPCIAs by practice #845.

Table 4 Descriptive statistics of the number of Defined Daily Doses (DDDvet) per cow and year for antimicrobial products used to treat acute mastitis on 232 Austrian dairy farms, based on European Medicines Agency technical units.

Treatment	ATCVet Code	Antimicrobial class	No. herds treating cows	% (N = 232)	nDDDvet/cow/year	
					Mean	SEM	Median	Minimum	Maximum	
INTRAMAMMARY										
Amoxicillin (& clavulanic acid)	QJ51CR02	Aminopenicillin	2	0.9	0.18	0.05	0.18	0.11	0.26	
Ampicillin & cloxacillin	QJ51CR50	Aminopenicillin	63	27.2	0.44	0.05	0.30	0.01	2.56	
Penicillin	QJ51CE09	Penicillin	83	35.8	0.63	0.08	0.41	0.08	4.19	
Cefalexin	QJ51DB01	Ceph. (1st gen.)	4	1.7	0.35	0.11	0.35	0.09	0.59	
Cefalexin & kanamycin	QJ51RD01	Ceph. (1st gen.)/aminoglycoside	28	12.1	0.60	0.10	0.51	0.07	2.61	
Cefazolin	QJ51DB04	Ceph. (1st gen.)	3	1.3	0.25	0.13	0.13	0.06	0.56	
Cefoperazone	QJ51DD12	Ceph. (3rd gen.)	89	38.4	0.60	0.07	0.35	0.03	2.96	
Cefquinome	QJ51DE90	Ceph. (4th gen.)	80	34.5	0.54	0.10	0.29	0.04	6.65	
Lincomycin & neomycin	QJ51RF03	Lincosamide & aminoglycoside	2	0.9	0.24	0.06	0.24	0.16	0.32	
Sulfadiazine & trimethoprim	QJ51RE01	Sulphonamide & trimethoprim	7	3.0	0.02	0.00	0.01	0.01	0.03	
SYSTEMIC										
Amoxicillin	QJ01CA04	Aminopenicillin	24	10.3	0.10	0.01	0.10	0.02	0.23	
Ampicillin	QJ01CA01	Aminopenicillin	9	3.9	0.09	0.01	0.07	0.04	0.14	
Penicillin	QJ01CE	Penicillin	99	42.7	0.16	0.02	0.11	0.01	1.01	
Benzylpenicillin & streptomycin	QJ01RA01	Penicillin & aminoglycoside	13	5.6	0.16	0.03	0.13	0.02	0.45	
Cefquinome	QJ01DE90	Ceph. (4th gen.)	62	26.7	0.15	0.02	0.09	0.01	1.11	
Ceftiofur	QJ01DD90	Ceph. (3rd gen.)	23	9.9	0.08	0.01	0.05	0.01	0.30	
Enrofloxacin	QJ01MA90	Fluoroquinolone	20	8.6	0.12	0.02	0.07	0.03	0.46	
Kanamycin	QJ01GB04	Aminoglycoside	15	6.5	0.03	0.00	0.03	0.01	0.07	
Marbofloxacin	QJ01MA93	Fluoroquinolone	57	24.6	0.13	0.02	0.08	0.02	0.82	
Oxytetracycline	QJ01AA06	Tetracycline	4	1.7	0.04	0.01	0.05	0.02	0.07	
Sulfadimidine & trimethoprim	QJ01EW03	Sulphonamide & trimethoprim	14	6.0	0.16	0.02	0.14	0.04	0.34	
Tylosin	QJ01FA90	Macrolide	53	22.8	0.13	0.02	0.09	0.01	0.59	
Notes.

nDDDvet/cow/year number of defined daily doses per cow and year

SEM standard error of the mean

ceph. cephalosporin

Discussion

The study presented here collated 6,334 observations on veterinary antimicrobial treatments on 248 Austrian dairy farms over a one-year period, from 17 veterinary practices and covering the equivalent of 6,467 cow-years. A total of 2,304 antimicrobial treatments in this study population were for udder diseases (excluding DCT). The current study went further than the statutory framework, which collates dispensing data only, and collected AMC data from dairy farms for antimicrobial substances both dispensed to farmers and those administered by veterinarians. With respect to AMC data, direct comparisons with previous studies are known to be challenging, due to the wide variety of study designs, AMC data collection and calculation methods used (Bondt et al., 2013; Merle et al., 2014; Postma et al., 2015; Collineau et al., 2017). For this reason, it is hoped that the introduction of standardised European technical units (Defined Daily Doses, DDDvet), as published by the EMA in 2016 and used in the present study, to calculate the number of DDDvet per cow and year, will help to harmonise AMC data reporting in future.

It is generally accepted that good quality data with respect to antimicrobial treatments in the livestock sector are difficult to obtain (Grave et al., 1999; Chauvin et al., 2001; Menéndez González et al., 2010). A number of authors have also reported that veterinary data can be assumed to be more complete than records kept by farmers (Menéndez González et al., 2010; Kuipers, Koops & Wemmenhove, 2016). In Austria, the reporting of electronic data on antimicrobials dispensed for use in food-producing animals became a legal requirement for veterinarians in 2015 and, for this reason, the data on both antimicrobial dispensing and administration collated here can be assumed to be close to complete and accurate. Where paper records were required, this was due to the current study’s additional requirement to collect data on antibiotics administered by the veterinarian and the need to analyse information by diagnosis. In the authors’ opinion, this is an advantage of the current study compared to previous studies, which have often relied on self-reporting by farmers and/or veterinarians. Furthermore, due to the long data collection period (one calendar year), the authors believe it is also unlikely that the participating veterinarians would have adjusted their reported antimicrobial prescribing behaviour due to their involvement in the study.

Figure 4 Proportions of defined daily doses by cow and year for non-critical versus HPCIA antimicrobial use for diagnoses of “udder disease” over a one-year period, by veterinary practice.

HPCIA, highest priority critically important antimicrobials.

Nevertheless, it is important to note that the relevance of this study may be limited by the fact that it was based on a convenience sample and therefore the results should be treated with caution when extrapolating them to relate to the whole of Austria. The authors acknowledge that the participating veterinarians were likely to be particularly interested in antimicrobial use and resistance and the likelihood of including unmotivated colleagues was low. However, in the current political climate, the collation of AMC data and diagnosis reporting are considered sensitive data requiring a considerable level of trust between veterinarians, farmers, and researchers and, for this reason, randomisation seemed unlikely to succeed in collecting adequate data (Collineau et al., 2017). A number of previous studies of antimicrobial consumption in cattle production have also used convenience samples and a Swiss study that attempted to recruit volunteers nationwide actually found that they enrolled a seemingly more progressive subset of farmers, with larger, more technologically advanced farms than the Swiss average (Menéndez González et al., 2010; Saini et al., 2012; Stevens et al., 2016). It is also important to note that all veterinarians and dairy farmers enrolled in this study were members of the Austrian Animal Health Service.

While focusing on udder disease, this analysis did not include dry cow therapy, as the EMA has not published defined daily doses (DDDvet) values for DCT. The European Medicines Agency has applied the “Defined Course Dose, DCDvet”, equivalent to four treatments per udder, to dry cow therapy and the authors felt that including this unit in the current study would have caused confusion when comparing therapeutic udder disease treatments with DCT. Furthermore, it is worth noting that concentrating on the therapeutic treatment of udder disease alone meant that only 36.4% of all observations of antimicrobial treatment in dairy cows collected over the one-year period were included in this study.

Although the sample population of the current study was not randomised, a comparison with national dairy cattle demographics by breed shows that the sample included here appears to be broadly representative of the Austrian national herd. The vast majority (73.8%) of the cows included in the study were of the Simmental (Austrian: Fleckvieh) breed, which corresponds almost exactly with the proportion of this breed (73.3%) among dairy cows enrolled in the national milk recording scheme (ZAR, 2016). Similarly, Brown Swiss (Braunvieh) cows made up 10.9% of the study population and 12.0% of the national herd, while Holstein-Friesians were 12.7% of the sample and 11.7% of the total dairy cow population (ZAR, 2016).

The European Medicines Agency uses the “population correction unit” (PCU) based on livestock demographics (including slaughter and animal movement data) to estimate the total weight of the livestock population in each country and compares veterinary AMC across the EU in mg/PCU in the ESVAC report (EMA, 2016a). Antimicrobial use in dairy cattle in Europe has often been reported to be much lower than in other livestock species and production systems (Merle et al., 2012; Merle et al., 2014; EMA, 2016a; Obritzhauser et al., 2016). In Austria, recent data (statutory reporting) on the amount of antimicrobials dispensed for use in cattle have shown that a total of 15.3 mg/PCU is used for this species (both dairy and beef production systems), compared to 79.2 mg/PCU for pigs (Fuchs & Fuchs, 2016). Of the total antimicrobial tonnage dispensed for use in cattle in Austria in 2015, 36.0% were for use in dairy cattle, 47.7% for use in beef production (including veal calves), 4.2% for use in breeding enterprises, and 12.0% in “other” production systems (i.e., a combination of dairy, beef and breeding systems) (Fuchs & Fuchs, 2016).

Numerous European surveys have demonstrated that treatment for udder disease is the single most common diagnosis leading to AMC in dairy cattle (Menéndez González et al., 2010; Merle et al., 2014; De Briyne et al., 2014; Kuipers, Koops & Wemmenhove, 2016). The present study confirmed this with 36.4% of all antimicrobial treatments being for diagnoses classed as “udder disease” (excluding DCT), compared to 22.9–26.5% to treat mastitis in the Netherlands between 2005 and 2012, and 35% for “udder and teat” treatments in Switzerland (Menéndez González et al., 2010; Kuipers, Koops & Wemmenhove, 2016). In Austria, intramammary treatments have also been shown to be the most common application form of antimicrobials in dairy cows in a previous observational study on 465 dairy farms between 2008 and 2010 (Obritzhauser et al., 2016). When calculated as the number of defined daily doses, the mean number of annual animal defined daily doses for acute mastitis in the present study was 1.21 DDDvet /cow/year (systemic and intramammary treatments), compared to intramammary treatment rates for clinical mastitis of 2.02 DDD/cow/year in the US, approximately 2.30 DDD/cow/year in Belgium and approximately 1.28 DDD/cow/year in Canada (Pol & Ruegg, 2007; Saini et al., 2012; Stevens et al., 2016). Furthermore, the number of defined daily doses per cow and year for udder disease as a whole was 1.33 DDDvet/cow/year in the present study (for both intramammary and systemic treatments), which is lower than recently reported solely for intramammary udder disease treatments in Ireland (1.4 DDDvet/cow/year) (More, Clegg & McCoy, 2017).

In the current study, systemic penicillins were used at a mean number of 0.16 (±0.02) DDDvet/cow/year and via the intramammary route at a mean DDDvet/cow/year of 0.63 (±0.08). Intramammary first generation cephalosporins combined with aminoglycosides were used at a mean daily dose rate of 0.60 DDDvet/cow/year in the present study, compared to a rate of approximately 0.82 DDD/cow/year in Belgium (Stevens et al., 2016).

The most commonly used intramammary HPCIA treatments for acute mastitis were the third generation cephalosporin, cefoperazone, in 38.4% of herds and the fourth generation cephalosporin, cefquinome, in 34.5% of herds. With respect to defined daily doses, cefoperazone was used at a mean of 0.60 (±0.07) DDDvet/cow/year, with cefquinome daily doses being slightly lower at a mean DDDvet/cow/year of 0.54 (±0.10). By comparison, Stevens et al. (2016) determined that a substantially lower level of third generation cephalosporins was used to treat clinical mastitis via the intramammary route at 0.37 DDD/1,000 cow-days (approximately 0.14 DDD/cow/year) in Belgium, and an even lower level was reported in Canada at 0.09 DDD/1,000 cow-days (approximately 0.03 DDD/cow/year) (Saini et al., 2012). The level of intramammary fourth generation cephalosporins used in the Belgian study was, however, virtually identical to that determined in the present study, namely 1.58 DDD/cow/1,000 cow-days (approximately 0.58 DDD/cow/year) (Stevens et al., 2016). Cefquinome (a fourth generation cephalosporin) was administered systemically for the treatment of acute mastitis in 26.7% of herds in the present study. The third generation cephalosporin, ceftiofur, was used systemically in 9.9% of herds, which contrasts favourably with a US study where third generation cephalosporins were used in 30% of herds (Pol & Ruegg, 2007). A further North American study reported six of 33 (18%) dairy farms using ceftiofur systemically to treat clinical mastitis, even though this was classified as off-label use according to the US licence at the time (Sawant, Sordillo & Jayarao, 2005). In the present study, ceftiofur was administered systemically at a mean daily dose rate of 0.08 DDDvet /cow/year, compared to 0.47 DDD/cow/year in the US (Pol & Ruegg, 2007).

Other HPCIAs, namely the macrolide, tylosin, and the fluoroquinolone, marbofloxacin, were administered systemically to treat udder disease in 22.8% and 24.6% of herds, respectively, both at a defined daily dose of 0.13 (±0.02) DDDvet/cow/year. These classes were not administered via the intramammary route in the current study as no such products are currently licensed for use in Austria. Colistin, a recent addition to the WHO’s HPCIA list, was not administered to calves or youngstock on any of the dairy farms included in the present study. This HPCIA is not licensed for use in dairy cows in Austria. The present study highlighted the importance of the individual prescribing veterinarian or practice with respect to the use of antimicrobials considered critically important for human health, with HPCIA use for the treatment of udder disease varying from <10% to >80% of the total number of defined daily doses administered by veterinary practice per cow and year.

Conclusions

The study presented here determined that dairy cattle in the study population in Austria were treated with antimicrobial agents at a relatively low and infrequent defined daily dose rate. The most frequently used antimicrobial group with respect to mastitis treatments was the beta-lactams, primarily penicillins, with third and fourth generation cephalosporins the most commonly used highest priority critically important antimicrobials with respect to both the proportion of herds treated and the number of defined daily doses administered per cow and year.

The authors would like to thank all the veterinarians and farmers who participated in the ADDA project and supported the research published here. Furthermore, we would like to acknowledge the support of the various ADDA project partners involved in all aspects of this research project, as well as Christian Laubichler, Dr Beate Pinior and Dr Sabine Hutter for their assistance in data analysis.

Additional Information and Declarations

Competing Interests

Author Contributions

Animal Ethics

Data Availability

The authors declare there are no competing interests.

Clair L. Firth conceived and designed the experiments, performed the experiments, analyzed the data, wrote the paper, prepared figures and/or tables, reviewed drafts of the paper.

Annemarie Käsbohrer and Corina Schleicher analyzed the data, contributed reagents/materials/analysis tools, reviewed drafts of the paper.

Klemens Fuchs conceived and designed the experiments, analyzed the data, contributed reagents/materials/analysis tools, reviewed drafts of the paper.

Christa Egger-Danner conceived and designed the experiments, performed the experiments, analyzed the data, contributed reagents/materials/analysis tools, reviewed drafts of the paper.

Martin Mayerhofer performed the experiments, analyzed the data, contributed reagents/materials/analysis tools, reviewed drafts of the paper.

Hermann Schobesberger and Josef Köfer conceived and designed the experiments, reviewed drafts of the paper.

Walter Obritzhauser conceived and designed the experiments, performed the experiments, analyzed the data, contributed reagents/materials/analysis tools, prepared figures and/or tables, reviewed drafts of the paper.

The following information was supplied relating to ethical approvals (i.e., approving body and any reference numbers):

The study was discussed and approved by the institutional ethics and animal welfare committee of the University of Veterinary Medicine, Vienna (Ref. No. ETK-13/11/2015), in accordance with good scientific practice guidelines and national legislation.

The following information was supplied regarding data availability:

The authors do not own the antimicrobial use data. These data were collected from the veterinarians who were treating the animals included in the study. Under the data privacy agreement signed by the farmers and veterinarians, these data are not available to be published, but were provided to the reviewers of this manuscript.

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
