# Peer review of "Antimicrobial consumption on Austrian dairy farms: an observational study of udder disease treatments based on veterinary medication records"

_PeerJ, doi:10.7717/peerj.4072_

## Round 0.1 · original submission · Minor Revisions

· Academic Editor

Minor Revisions

Two experts in the field have reviewed your manuscript and both suggested that your paper may be acceptable to PeerJ once minor revisions were undertaken.

Both reviewers highlight issues with the reference style, and this should be carefully addressed.

Reviewer 1 makes the point that the paper should address the limitations of your study design, and therefore the inferences made therein, and should be addressed near the start of the discussion (I note you have made reference already to limitations, therefore editing/reorganising is required). I also agree with reviewer 1 that the MS might benefit from some shortening, and concentrating the major findings.

Please explicitly explain all abbreviations within the MS as suggested by Dr. Botha, and clarify your comparisons with the New Zealand study.

Reviewer 1 ·

Basic reporting

In general my opinion is that the manuscript is written in an extensive way. There is a lot information that is actually not really needed for the reader. I believe that the more interesting information that is presented in this study would/could be more convincing when it is presented as short communication. In its current version it contains too much side information that distracts the reader from the essential and good/nice results of the study. For instance, the introduction and discussion are rather long. The results focus quite extensively on the number of treating farms, while I believe the essential information is included in the nDDDvet results. Furthermore, the statistical comparison between acute and chronic treatments is nice, but in my opinion not necessary information.


L12-13: 89.4% and 10.8% does not sum up to 100%
Some sentences are formulated in a strange/difficult way, e.g. L27 “the highest mean number”. I suggest that the manuscript is checked by a native speaker. E.g. L262-263. I believe you mean “6334 single antimicrobial treatments of dairy cows were observed…”
L43 these products are actually “antibiotic-free”, since there is a withdrawal period. Consider rephrasing
L93 “population correction unit” instead of “corrected”
L183 per prescribing “veterinarian” or “practice”? The results are presented per veterinary practice.
L194 the multiplication sign “x” should not be in italic
L230 drop “the calculation of”
L268 drop “As would be”
L285-286 this information is not necessary in the text, below the table is sufficient.
L305-309 please rephrase, the test and its properties were already mentioned in the M&M section, so no need to repeat them here.
L310-314 Please rephrase, this is very technical and not clear what you mean. The sentence is 5 lines long and the message is unclear to me.
L327 there were no 6334 datasets. There were 6634 observations (= single antimicrobial treatments) in the dataset…
L350 “other” do you have an example of this?
L376 drop “that”
Please check the references. E.g. L523 Preventive Veterinary Medicine with capitals.

Experimental design

The study design is well described. However the authors should discuss the disadvantages of their design more thoroughly (see validity of the findings).

Validity of the findings

The discussion should stress more the external validity of these results. Although it is already present in the discussion, I would start the discussion with the novelty of the results (as actually present) and the drawbacks of the study design (currently mentioned at the end of the discussion):
- Convenience sampling
- No dry cow therapy
- Only 30.6% of AM treatments in dairy cattle used
- L216-217. Although this will probably only be a minority of the treatments, instead of not including these treatments, you could consider to estimate the AMC of oxytetracycline sprays, in analogy with Postma et al., 2015: “…1 mL will be sprayed per second and the fact that 3 s of spraying is the average duration mentioned in the SPCs for topical spray product…”
How do all these drawbacks influence the comparison with other studies?

·

Basic reporting

Basic reporting and the use of English are good.
The authors explained the context of antimicrobial consumption in Austria well, and used relevant literature.
There are a few things that have to be addressed.

The are many acronyms in the article that are not explained, for example TGD (line 62 - it is explained in line 131), OIE (line 79), LKV (line 132),RDV (line 152), ESVAC (line 338), EDC, EFSA (line 423).

Line 207; "...., rather than per kilogram." (I assume you are referring to live weight?)

Lines 340-342. Are you referring to Germany (Merle et al), Austria (Obritzhauser et al), the EU, Europe or globally? Please state.

Lines 346-348 that refer to the New Zealand study: it is not only the calculation methods that distort the comparison, but in Austria dairy and beef production systems have been combined, while the NZ findings refer to dairy farms only. This makes a big difference and should be stated in the paper, or the NZ comparison should be deleted.

There seems to be an issue with one reference. Line 548. As it is, the authors have a mix of first and second joint report, and I believe the reference should be changed.
In APA 6th format it is:
ECDC (European Centre for Disease Prevention Control), EFSA (European Food Safety Authority), & EMA (European Medicines Agency). (2015). ECDC/EFSA/EMA first joint report on the integrated analysis of the consumption of antimicrobial agents and occurrence of antimicrobial resistance in bacteria from humans and food-producing animals. EFSA Journal, 13(1), 4006-4120. doi: 10.2903/j.efsa.2015.4006
In HARVARD is should be:
ECDC (EUROPEAN CENTRE FOR DISEASE PREVENTION CONTROL), EFSA (EUROPEAN FOOD SAFETY AUTHORITY) & EMA (EUROPEAN MEDICINES AGENCY) 2015. ECDC/EFSA/EMA first joint report on the integrated analysis of the consumption of antimicrobial agents and occurrence of antimicrobial resistance in bacteria from humans and food-producing animals. EFSA Journal, 13, 4006-4120.

Experimental design

Experimental design is sound and the method is described well.
The limitations of the research have been well described.
Using the standardised "Defined Daily Doses" for veterinary medicine of the EMA and use of AMC of active substances is a useful approach.
I believe the paper is within the scope and aims of the journal.

Validity of the findings

The findings are a good benchmark for Austria and a good comparison and contrast with other EU states.

Additional comments

I have gone through what the previous referees have said and I believe you have responded well to their comments.

---

## Round 0.2 · accepted · Accept

· Academic Editor

Accept

I am satisfied that you have addressed all the concerns raised by the reviewers, and happy to accept your paper for publication with PeerJ.

Kind regards

Andrew